# Transcriptomic Analysis of mRNA Expression Profiles in the Microglia of Mouse Brains Infected with Rabies Viruses of Varying Virulence

**DOI:** 10.3390/v15061223

**Published:** 2023-05-23

**Authors:** Jundan Liu, Wangchang Li, Dongling Yu, Rong Jin, Hualin Hou, Xiaoqing Ling, Abraha Bahlbi Kiflu, Xiankai Wei, Xiaogan Yang, Xiaoning Li, Yongming He, Ting Rong Luo

**Affiliations:** 1State Key Laboratory for Conservation and Utilization of Subtropical Agro-Bioresources, Guangxi University, Nanning 530004, China; liujundan@st.gxu.edu.cn (J.L.); liwangchang1019@163.com (W.L.); 18037162780@163.com (H.H.); 18376698042@163.com (X.L.); kifluabraha88@163.com (A.B.K.); xiaoningli@gxu.edu.cn (X.L.); 2College of Animal Sciences and Veterinary Medicine, Guangxi University, Nanning 530004, China; 3Guangxi Key Laboratory of Animal Breeding, Disease Control and Prevention, Nanning 530004, China; wxkdnnm@163.com; 4Guangxi Zhuang Autonomous Region Engineering Research Center of Veterinary Biologics, Nanning 530004, China; 5School of Life Science and Engineering, Foshan University, Foshan 528225, China

**Keywords:** rabies virus, microglia, molecular mechanisms, transcriptomic analysis

## Abstract

Rabies is a lethal encephalitis caused by the rabies virus (RABV) with a fatality rate near 100% after the onset of clinical symptoms in humans and animals. Microglia are resident immune cells in the central nervous system. Few studies have been conducted on the functional role of microglia in RABV infection. Here, we performed a transcriptomic analysis of mRNA expression profiles in the microglia of mouse brains intracerebrally infected with RABV. We successfully isolated single microglial cells from the mouse brains. The survival rate of dissociated microglial cells was 81.91%–96.7%, and the purity was 88.3%. Transcriptomic analysis revealed 22,079 differentially expressed mRNAs identified in the microglia of mouse brains infected with RABV strains (rRC-HL, GX074, and CVS-24) of varying virulence at 4 and 7 days post-infection (dpi) compared to the control group. The numbers of DEGs versus the control at 4 and 7 dpi in mice infected with rRC-HL, GX074, and CVS-24 were 3622 and 4590, 265 and 4901, and 4079 and 6337. The GO enrichment analysis showed that response to stress, response to external stimulus, regulation of response to stimulus, and immune system process were abundant during RABV infection. The KEGG analysis indicated that the Tlr, Tnf, RIG-I, NOD, NF-κB, MAPK, and Jak-STAT signaling pathways were involved in RABV infection at both 4 and 7 dpi. However, some phagocytosis and cell signal transduction processes, such as endocytosis, p53, phospholipase D, and oxidative phosphorylation signaling pathways, were only expressed at 7 dpi. The involvement of the Tnf and Tlr signaling pathways prompted us to construct a protein–protein interaction (PPI) network of these pathways. The PPI revealed 8 DEGs, including Mmp9, Jun, Pik3r1, and Mapk12. Notably, Il-1b interacted with Tnf and Il-6 with combined scores of 0.973 and 0.981, respectively. RABV causes significant changes in mRNA expression profiles in the microglia in mice. 22,079 differentially expressed mRNAs were identified in the microglia of mice infected with RABV strains of varying virulence at 4 and 7 dpi. The DEGs were evaluated using GO, KEGG, and PPI network analysis. Many immune pathways were up-regulated in RABV-infected groups. The findings will help elucidate the microglial molecular mechanisms of cellular metabolism dysregulated by RABV and may provide important information for investigating RABV pathogenesis and therapeutic methods.

## 1. Introduction

Rabies virus (RABV) is a highly neurotropic virus that inevitably causes lethal neuroencephalitis in mammals after the onset of neurological signs [1]. Every year, over 70,000 people die of rabies worldwide; thus, rabies poses a serious threat to public health [2]. RABV, belonging to the *Lyssavirus* genus of the *Rhabdoviridae* family, is a non-segmented negative-stranded RNA virus. The 12 kb RABV genome comprises five open reading frames encoding five viral structural proteins, including nucleoprotein (N), phosphoprotein (P), matrix protein (M), glycoprotein (G), and large RNA polymerase (L). All five proteins are essential for viral replication and spread [3].

Microglia are resident immune cells in the central nervous system (CNS). Microglial cells comprise 10%–15% of all CNS cells. These microglia are primarily brain-residing immune cells, maintaining CNS homeostasis function through phagocytic activity, and removing pathogens and cell detritus [4]. Microglia are usually activated by infection, neuronal injury, and inflammation after a CNS injury [5].

RABV stimulates CNS immune responses and causes dysfunction; the virus induces cytokine, interleukin, and chemokine synthesis and release, for example, stimulating Cxcl10 and Ccl5 release by activating protein kinase and NF-κB signaling pathways [6]. With the development of biotechnology, high-throughput sequencing technology has been applied to brain tissue from mice or cells infected with RABV [7,8,9]. Ji et al. applied microarray technology to analyze the long non-coding RNA expression in human neuroblastoma cells infected with RABV [10]. Kim et al. reported early transcriptional changes in the BV2 microglial cell line and neurons infected with RABV in vitro. He found that RABV infection impeded the spontaneous activity of neurons, and rabies virus-associated pathways were involved in neuronal differentiation, synaptic activity, and morphological changes [11]. In addition, Masuda et al. revealed that distinct microglial clusters exist in neurodegenerative and demyelinating diseases through single-cell transcriptomics studies [12]. Jordao et al. utilized single-cell profiling to identify subsets of myeloid cells with distinct fates during neuroinflammation [13]. Recently, Zhang et al. explored the use of fluorescence micro-optical sectioning tomography (fMOST) and single-cell RNA sequencing (scRNA-seq) to map the spatial and cellular distribution of RABV in the whole brain [14].

Our previous study revealed increased microglial numbers in brains infected with RABV. However, it remains unknown whether mRNA expression levels in microglia are affected by RABV. In this study, we applied transcriptomic analysis to identify the differentially expressed mRNAs in the microglia of mouse brains infected with RABV strains of varying virulence. We analyzed the potential changes in functions, interactions, and signaling pathways associated with the identified differentially expressed mRNAs and demonstrated that numerous proinflammatory cytokines (such as Tnf-α, IL-1β, and iNOS), interleukins, and chemokines were induced in the brain microglia of mice infected with RABV. Our findings will explain pathogenic mechanisms in RABV-infected microglial cells and provide novel insights into the role of differentially expressed mRNAs in the host immune response against RABV infection.

## 2. Materials and Methods

### 2.1. Viruses

The rRC-HL strain was successfully rescued by reverse genetics from the infectious cDNA of the RC-HL strain used to produce animal vaccines in Japan, which was donated by Professor Nobuyuki Minamoto (Gifu University, Japan) [15]. The GX074 RABV strain was isolated from a healthy-appearing dog brain from Debao County, Baise City, Guangxi Province of Southern China. The mouse-adapted rabies virus strain CVS-24 was provided by Professor Ling Zhao of Huazhong Agricultural University, China.

### 2.2. Virus Titration

Immunocytofluorescence (IF) was performed as previously reported to determine virus titers in cells [16,17]. The titers of rRC-HL in 6 passages were determined as 3.967 × 10^7^ focus-forming unit per milliliter (FFU/mL) in N2a cells by IF. The 50% lethal dose (LD_50_) of GX074 in 6 passages was calculated as 10^−4.898^/mL in adult mice [18], and that of CVS-24 in 8 passages was 10^−5.843^/mL according to Reed-Muench [19]. Briefly, 50 μL rRC-HL was serially diluted 10 times, and 100 μL of each dilution was inoculated into N2a cells in 96-well plate monolayer cells. After incubating RABV on N2a cells at 37 °C for 1 h, we removed the virus inoculum and added 100 μL semi-solid medium containing 2% FBS. The cells were then cultured at 37 °C for 48 h, fixed with ice-cold acetone: methanol (1:1) and stained with FITC-conjugated anti-RABV N protein antibody (Hangzhou Dayao Biotechnology Co., Hangzhou, China, Ab0056). Fluorescent foci were subsequently counted under a Nikon IX51 fluorescence microscope (Nikon, Tokyo, Japan), and the virus titer was calculated as FFU/mL.

### 2.3. Mouse Infection Test

Four-week-old Kunming (KM) mice were purchased from Hunan Saike Jingda Experimental Animal Co. Ltd. (Changsha, China). The mice were raised in a biosafety laboratory under controlled humidity, temperature, and light (12/12 h light/dark cycle) with free access to food and water. The mice were randomly assigned to four groups, six per group, half male and half female. A dilution series of virus stocks was prepared using DMEM in an ice bath. For the virus-infected group, mice were inoculated intracerebrally with 1000 FFU of rRC-HL in 30 µL DMEM. The mice were challenged with CVS-24 and GX074 at 100 LD_50_ in 30 µL DMEM, and the control group was inoculated with 30 µL DMEM.

The mice were monitored for clinical manifestations three times daily for 7 days post-infection (dpi); any death within the first 3 dpi was considered non-specific. Disease progression was assessed by scoring clinical signs and body weight as previously described [20]. The mice were then euthanized at 4 dpi and 7 dpi using halothane in a closed container.

KM mice were intracerebrally infected with RABV strains with varying virulence. Following RABV infection, the virus replicates in the brain and triggers a strong host immune response and neuroimmune inflammation. The initial clinical signs (the beginning of symptoms) were observed in mice at 4 dpi with neuro-manifestations; mice infected with RABV showed slight clinical signs at 4 dpi, i.e., hunched back, depressed spirit, and disordered hair. And the mice showed tremors and paralysis of the fore or hind limbs, hypothermia, abundant white discharge from the eyes, and respiratory failure at 7 dpi. The mice were sacrificed at 4 dpi and 7 dpi, and brains were collected for microglial separation by magnetic beads separation (MACS). Then, isolated microglia were assessed by transcriptomic RNA-seq analysis or real-time quantitative PCR (RT-qPCR).

### 2.4. Brain Sample Collection and Microglia Dissociation

Mouse brains at 4 dpi and 7 dpi were collected. Each brain sample was washed 5 times with PBS, and the cerebral meninges and blood vessels were removed by ophthalmic forceps. The olfactory bulb, cerebrum, and cerebellum were dissected using microscissors and microtweezers on a dissecting plate, and the mouse brain was roughly chopped into fine pieces using scissors [21,22]. Enzyme cocktails were added to the brain tissue to disperse the cells using an adult brain dissociation kit (Miltenyi, Bergisch Gladbach, Germany, 130-107-677) according to the manufacturer’s instructions. Then, the microglia were positively selected using an antigen–antibody-mediated magnetic cell sorting assay via CD11b Microglia MicroBeads (Miltenyi, Bergisch Gladbach, Germany, 130-093-634). Briefly, the mixed cells were re-suspended in PBS containing 0.5% FBS. The cell suspension was processed using a Mini MACS Starting Kit (Miltenyi, Bergisch Gladbach, Germany, 130-090-312). The unlabeled cells were allowed to pass through the MS (Mini Separation) column, which captured the labeled cells. The column was washed with PBS, removed from the magnetic separator to a suitable tube, and flushed with PBS to collect the purified microglia population.

### 2.5. Microglia Purity and Morphological Analysis

Microglia purity was assessed by IF and flow cytometry. Microglial cells were fluorescently stained with CD11b^-PE^ (BD, Franklin Lakes, NJ, USA, 553311) and CD45^-FITC^ (BD, Franklin Lakes, NJ, USA, 553080). Cell debris and dead cells were excluded from the analyzed microglia using scatter signals and 7-AAD (BD, Franklin Lakes, NJ, USA, 559925) staining solution. For morphological analysis, microglial cells were fixed in 4% formaldehyde for 15 min, permeabilized with 1% TritonX-100, and blocked with 5% BSA. The primary antibody, Iba-1 (FUJIFILM Wako Chemicals USA, Corp, Richmond, VA, USA 019-19741, 1:1000 dilution), a monoclonal antibody specific to microglia, was added, and the cells were incubated overnight at 4 °C. The secondary antibody, goat anti-rabbit IgG (H + L) (Invitrogen, Waltham, MA, USA, A-11011, 1:1000 dilution), was subsequently added, and the cells were incubated for one hour at 37 °C. Images were captured using a NIS-Elements AR3.2 imaging system (Nikon, Tokyo, Japan).

### 2.6. RNA Extraction, cDNA Library Construction, Sequencing and Data Processing

Total RNA was extracted from microglia using TRIzol reagent (Invitrogen, Waltham, MA, USA, 15596026), following the manufacturer’s instructions in our experiments. Shanghai Personal Biotechnology Cp. Ltd. (Shanghai, China) assessed the RNA quality and quantity by the absorbance at 260 nm/280 nm using a NanoDrop ultraviolet spectrophotometer (Thermo Scientific, Waltham, MA, USA); RNA integrity was verified by 1% agarose gel electrophoresis. The cDNA library was purified on the AMPure XP system (Beckman Coulter, Beverly, CA, USA), and the library was sequenced on a NovaSeq 6000 platform (Illumina, San Diego, CA, USA). RNA-sequencing reads were first trimmed to remove poly(A) and unqualified reads with Cutadapt (v1.15), then aligned to the *Mus musculus* GRCm38 genome using Ensembl. The counts were summarized at the gene level using HTseq (0.9.1). Gene expression values were calculated from fragments per kilo basis per million fragments (FPKM). These FPKM values were used to generate a heatmap with the heatmap R package. Paired differential gene expression analyses were performed with DEseq (1.30.0) with screened conditions as follows: multiple expression difference|log2fold change| > 1 and *p*-value < 0.05. According to the expression level of the same gene in different samples and the expression patterns of different genes in the same sample, heatmaps were generated with the Euclidean method to calculate the distance and the Complete Linkage method for clustering.

GO and KEGG enrichment analyses of differential genes were performed using topGO and ClusterProfiler (3.4.4) software, respectively. The main biological functions associated with the differentially expressed genes were determined by GO (*p*-value < 0.05). The KEGG pathway enrichment analysis of the differential genes focused on the associated enriched pathways (*p*-value < 0.05).

A protein–protein interaction (PPI) analysis of differential genes was performed using the STRING database (http://string-db.org (accessed on 15 August 2022)) to reveal the relationships between the target genes. The PPI network model was generated using Cytoscape [23,24].

### 2.7. RT-qPCR for mRNA Sequencing Validation

The twelve primer pairs (Table 1) were designed and obtained from NCBI databases (https://www.ncbi.nlm.nih.gov/ (accessed on 26 March 2021)). The primers were synthesized and purified by BGI Biotech Cp. Lid (Shenzhen, China). RNA was reverse-transcribed into cDNA using FastKing gDNA Dispelling RT SuperMix (TIANGEN, Beijing, China, KR118). RT-qPCR was performed at a final volume of 20 μL, including 10 μL Universal SYBR Green Supermix (ABconal, Woburn, MA, USA, RK21203), 1 μL forward primer, 1 μL reverse primer, 2 μL cDNA, and 6 μL H_2_O on a Roche LightCycler^®^ 96. The RT-qPCR cycling conditions were as follows: 95 °C for 3 min, followed by 45 cycles of 95 °C for 15 s and 60 °C for 30 s. The primers and sequences are listed in Table 1. Gene expression was normalized to β-actin. The expression levels of genes were measured in terms of threshold cycle values using the 2^−ΔΔCt^ method [25].

### 2.8. Bioinformatics Analysis

Differential expression was assessed using transcript abundances as DESeq (version 1.18.0) inputs. The transcripts with log2 fold change >1 and *p* < 0.05 were considered differentially expressed genes (DEGs). The cluster analyses of the DEGs were performed using R. A principal component analysis (PCA) was conducted on the DEGs per sample. The GO enrichment analysis was performed using top GO. The *p*-values were calculated using the hypergeometric distribution method to obtain the significantly enriched GO terms to identify the major biological functions associated with the DEGs. KEGG is a comprehensive data library that combines information on genomic, chemical, and system functions. The number of DEGs associated with different KEGG pathways was determined to identify the linked signaling pathways.

Data were statistically compared using a *t*-test between two groups and two-way ANOVA with Tukey’s multiple comparison within groups. The graphs were analyzed using GraphPad Prism 9.4.1 (GraphPad Software, Inc., San Diego, CA, USA). *p*-values < 0.05 were considered to indicate significance.

## 3. Results

### 3.1. Mouse Infection Test

The mice were intracerebrally infected with RABV strains rRC-HL, CVS-24, or GX074 in DMEM; a control group was inoculated with DMEM. The mice showed preliminary clinical signs at 4 dpi, and the typical neurological symptoms of rabies occurred at 7 dpi. The mice brains at 4 dpi and 7 dpi were subsequently collected, and microglia were isolated for transcriptomics analysis (Figure 1).

To investigate how RABV affects microglia in vivo, KM mice were intracerebrally infected with rRC-HL, GX074, and CVS-24 in DMEM; the control group was injected with the same volume of DMEM. Mice infected with rRC-HL lost weight, showed reduced movement at 4 dpi, and recovered at 7 dpi. Mice infected with CVS-24 showed weight loss (3 and 4 dpi), hair ruffling, loss of coordination (4–5 dpi), tremors, paralysis of the fore or hind limbs, and hind limb palsy (6–7 dpi). The mice infected with GX074 tended to show weight loss (starting from 5 dpi), mania, convulsions, trembling, shaking, and even paralysis (6–7 dpi). The mice infected with CVS-24 exhibited body weight decreasing significantly between 4 dpi and 7 dpi, and the body weight change was from 109.15% to 87.45% (*p* < 0.01). The body weight of mice challenged with GX074 was reduced from 5 dpi to 7 dpi, from 132.37% to 105.22% (*p* < 0.05). In contrast, the body weight of the mice infected with rRC-HL declined slightly, from 112.87% at 5 dpi to 100.74% at 7 dpi. The weight of mice inoculated with DMEM increased over time. The changes in mouse body weight are shown in Figure 1a. The rabies symptoms of CVS-24-infected mice appeared at 4 dpi, and typical neurological symptoms were observed at 6 and 7 dpi; the highest clinical score was 4, presenting hypothermia, abundant white discharge from the eyes, and respiratory failure. The mice infected with GX074 showed typical neurological symptoms at 5 dpi, which became severe at 7 dpi, presenting clinical scores ranging from 3 to 4 at 6 and 7 dpi, tremors, and paralysis of the fore or hind limbs. In contrast, the mice infected with rRC-HL had mild symptoms, occurring from 5 to 7 dpi with a score of 1 to 2 in the broken-line graph, showing disordered hair, slackness, and a slight depression (Figure 1b).

### 3.2. Separation and Identification of Microglia

Single microglial cells were dissociated from adult mouse brains using an adult brain dissociation kit and sorted by MACS [26]. First, we detected N gene mRNA expression in microglial cells by RT-qPCR to confirm RABV infection. The mRNA content of the N gene was increased from 4 to 7 dpi in all groups except for the DMEM control group; those infected with CVS-24 showed significant differences at 4 and 7 dpi compared with the DMEM control (*p* < 0.01) (Figure 1c). Then, we used trypan blue to identify the survival of the dissociated microglia. The minimum survival rate of dissociated microglia was 81.91% in mice infected with GX074 at 4 dpi, and the maximum was 96.7% in those infected with rRC-HL at 7 dpi. Significant differences were observed between the DMEM group at 4 dpi and the CVS-24 group at 4 dpi (DMEM-4d vs. CVS-24-4d), the DMEM group at 7 dpi and the GX074 group at 7 dpi (DMEM-7d vs. GX074-7d), and the DMEM group at 7 dpi and the CVS-24 group at 7 dpi (DMEM-7d vs. CVS-24-7d) (*p* < 0.05). GX074-4d vs. CVS-24-4d and DMEM-7d vs. rRC-HL-7d showed a strong significant difference (*p* < 0.01) (Figure 1d). We identified the purity of microglia by IF and flow cytometry [27,28]. Almost all the separated microglia were stained red with the specific anti-microglia MAb against Iba-1 (Figure 1e). Meanwhile, the FACS revealed that the microglial cells in the positive fraction were enriched to a purity of 88.39%. Before separation, the proportion of microglia in the brain was 13.47% [27], and 0.98% of the negative fraction comprised microglia (Figure 1f). Thus, microglia demonstrated good activity and purity for the follow-up experiments.

### 3.3. Determination of Differentially Expressed mRNAs in Microglia from RABV-Infected Brains

mRNA-seq was conducted to further understand the effects of RABV infection on the microglia transcriptome. As the results indicated, 43.07–56.14 million raw reads were obtained from the 24 samples using the Illumina Sequencing Platform. We then used Cutadapt (v1.15) to filter the sequencing data to obtain high-quality sequences (clean data) for further analysis. Quality control data for mRNA sequences, including Q20, Q30, and clean reads, are presented in Appendix A.

Pearson correlation coefficient analysis showed that the samples were divided into two clusters: one cluster included DMEM-4d, DMEM-7d, and GX074-4d (blue frame) with Pearson correlation coefficients exceeding 0.8, representing extremely strong positive correlations; the second cluster included rRC-HL-4d, CVS-24-4d, rRC-HL-7d, GX074-7d, and CVS-24-7d (green frame) with Pearson correlation coefficients also exceeding 0.8 but distinguished from the first cluster through a correlation coefficient below 0.6 (Figure 2a). The PCA of the complete microglial transcriptomes demonstrated an obvious distinction among the different groups. The DMEM-4d, DMEM-7d, and GX074-4d groups showed similar variance, as did the rRC-HL-4d, rRC-HL-7d, GX074-7d, and CVS-24-4d groups; only the CVS-24-7d group showed no correlation with any other group (Figure 2b). These results showed that the differentially expressed mRNAs in microglia induced by different RABV strains between 4 and 7 dpi in this study were reliable, and the characteristics of microglia cells infected with different RABV strains differed.

The DEG histogram showed more DEGs in microglial cells at 7 dpi than at 4 dpi in mice infected with the same RABV strain. The rRC-HL-4d vs. rRC-HL-7d ratio was 3741:6367, GX074-4d vs. GX074-7d was 266:2664, and CVS-24-4d vs. CVS-24-7d was 4079:4962 (Figure 2c). These data showed that RABV upregulates DEGs and continues to upregulate DEGs as the infection progresses.

RNA-seq analysis revealed obvious differences among the DEGs. We further analyzed significant DEGs (|log2 fold change| > 1, *p*-value < 0.05) between different comparison groups and constructed Venn diagrams and histograms. The upregulated numbers of DEGs in the DMEM-4d vs. rRC-HL-4d, DMEM-7d vs. rRC-HL-7d, DMEM-4d vs. GX074-4d, and DMEM-7d vs. GX074-7d groups were 260, 1290, 16, and 301, respectively. The down-regulation numbers of DEGs in these comparison groups, including DMEM-4d vs. CVS-24-4d and DMEM-7d vs. CVS-24-7d, rRC-HL-4d vs. GX074-4d and rRC-HL-7d vs. GX074-7d, and rRC-HL-4d vs. CVS-24-4d and rRC-HL-7d vs. CVS-24-7d, were 858:339, 196:187, and 300:24, respectively. Eighteen core genes were common across all comparison groups at 4 dpi and 42 at 7 dpi (Figure 3a,b).

We also analyzed the differential expression of cytokines in different groups. Overall, DMEM-4d, DMEM-7d, and GX074-4d-infected groups showed upregulated expression of Il-17c, Ccl6, Ptgds, Ccl9, Ccl24, Cxcl17, Il-16, and Ltc4s. Oas-, Ifn-, Il-, and Tnf-like factors were highly expressed in different RABV-infected groups. rRC-HL-4d and CVS-24-4d infected groups showed that RABV could stimulate interferons and Oas-like factors Oas1g, Oas1a, Oas2, and Oas3. Some interleukins and chemokines, such as Il-1b, Il-10, Il-27, Cxcl9, Ccl7, and Cxcl16, were highly expressed in the CVS-24-7d and GX074-7d groups. More cytokine genes were highly upregulated in the GX074-7d group (Figure 2d).

### 3.4. RT-qPCR Validation of mRNA Expression in Microglia

RT-qPCR was used to validate RNA-seq data by verifying DEGs in different groups at 7 dpi. Tnf and Tlr signaling pathways DEGs were selected for RT-qPCR analysis. The primers used for RT-qPCR are shown in Table 1. RT-qPCR validation showed that the mRNA expression levels of DEGs (e.g., Cxcl9, Stat1, and Il-1β) were highly upregulated in the rRC-HL group; Cxcl10, iNOS, and Sphk1 were highly expressed in the CVS-24 group; and Ikbke, IRF7, Tnf-α, and MCP-1 were upregulated in the GX074 group (Figure 4). These findings indicate that the changes in DEG mRNA expression were consistent with those obtained in transcriptomics analysis (Figure 4); thus, the DEG data were reliable.

### 3.5. Functional Prediction and Signaling Pathway Analysis via GO and KEGG Enrichment Analysis

All DEGs were mapped to each GO database term in the GO enrichment analysis. The significantly enriched terms were calculated by hypergeometric distribution with the whole genome as background. GO terms with significant DEG enrichment and related biological functions were obtained in the GO functional enrichment analysis. The results were divided into biological process, cell component, and molecular function. We identified significantly enriched 21 GO terms at 4 dpi and 19 at 7 dpi, which were the most significant enrichments to be analyzed. The results showed that response to stress, response to external stimulus, regulation of signaling, regulation of response to stress, regulation of response to stimulus, and immune system process were abundant in all comparison groups (DMEM vs. CVS-24, DMEM vs. GX074, DMEM vs. rRC-HL, rRC-HL vs. GX074, and rRC-HL vs. CVS-24) at 4 dpi and became stronger at 7 dpi. DMEM vs. GX074 showed less significant enrichment in GO terms at 4 dpi than at 7 dpi (red frame). However, rRC-HL vs. GX074 showed more significant enrichment at 4 dpi than at 7 dpi (black frame). The same trend was observed in the KEGG pathway analysis in DMEM vs. GX074 (red frame) and rRC-HL vs. GX074 (black frame) (Figure 5b), showing more active pathways at 7 dpi than at 4 dpi in DMEM vs. GX074. These results also indicate that RABV activates the immune response in microglia. Interestingly, the cellular metabolic process was enriched at 4 dpi (Figure 5a) and negative regulation of inflammatory response at 7 dpi (Figure 5b). These results indicated a more active cellular metabolic process induced by RABV infection at 4 dpi. As the disease progressed and the mice began to die at 7 dpi, the inflammatory response became weaker. DEGs involved in lipid raft, viral life cycle, and viral entry into host cells were not enriched in rRC-HL vs. CVS-24 and rRC-HL vs. GX074 at 7 dpi, suggesting that the process of RABV entry into the host cells occurred via the same pathway in CVS-24 and GX074 infection. Microglia are related to neurogenesis, neuron differentiation, and neuron development. In this study, microglia played a vital role in the death of neurons infected with RABV. The activated signaling pathways in mouse brains during RABV infection are displayed via GO term analysis in Appendix A.

We chose 19 signaling pathways activated by RABV at 4 dpi and 24 signaling pathways activated by RABV at 7 dpi for KEGG analysis. These signaling pathways showed high enrichment, demonstrating substantial differences among the RABV comparison groups (Figure 5c,d). Endocytosis, Ras-, p53-, mitophagy-, phospholipase D, and oxidative phosphorylation signaling pathways were mainly expressed at 7 dpi in different RABV comparison groups, demonstrating that phagocytosis and cell signal transduction played an important role in microglia infected with RABV. However, oxidative phosphorylation and the IL-17 signaling pathway were only present in DMEM vs. CVS-24 at 7 dpi. The immune system plays an important role in immune surveillance, defense, and regulation. Innate immunity is the basis of all immune responses. Tlr, Tnf, RIG-I, Nlr, NF-κB, endocytosis, cellular senescence, and antigen processing and presentation pathways were highly expressed at 4 dpi and 7 dpi in different RABV comparison groups except for rRC-HL vs. GX074 and rRC-HL vs. CVS-24 at 7 dpi, demonstrating that these signaling pathways were similar in GX074 and CVS-24 at 7 dpi. Among them, cytokine-cytokine receptor interaction showed higher expression at 4 dpi than at 7 dpi in DMEM vs. rRC-HL and DMEM vs. CVS-24, indicating that RABV could trigger innate immunity earlier. Ras and Rap1 signaling pathways played an important role in apoptosis, cell proliferation, and differentiation. Pathways related to nervous system diseases were also activated by RABV, including those associated with neurodegenerative diseases such as Huntington’s disease, Alzheimer’s disease, Parkinson’s disease, and glioma. These data are presented via KEGG analysis in Appendix A.

### 3.6. PPI Network Analysis

Microglial cells play an important role in inflammation in the CNS and can remove pathogenic microorganisms. We constructed a PPI network to identify interacting proteins specific to inflammation in microglia infected with RABV to develop an in-depth understanding of the correlation between microglia and inflammation. We extracted genes enriched in GO and KEGG analyses via STRING analysis and visualized the network using Cytoscape software (3.8.2) with a threshold interaction score of 0.70. We chose two pathways to perform an in-depth analysis to better understand the effect of RABV infection on microglia. First, Tnf signaling pathway and primary neighborhood interacting partner gene expression levels were shown. Some genes, such as Map3k5, Mmp9, Mapk12, Pik3r2, Mapk14, Creb312, Map2kb, and Jun from the Tnf signaling pathway, were upregulated in the DMEM-4d, DMEM-7d, and GX074-4d groups. The expression levels of Cxcl10, Ccl2, Bcl3, Cxcl2, Cxcl3, and Tnf, which regulate inflammatory responses, were also upregulated at 7 dpi (Figure 6a). Il-1β potentially interacts with Tnf and Il-6, with combination scores of 0.973 and 0.981, respectively (Appendix A).

Tlrs, which are pattern recognition receptors (PRRs) that recognize pathogen-associated molecular patterns (PAMPs), are conserved and specific molecules induced by pathogens. Tlrs ligation can trigger distinct but shared signaling pathways, leading to effector mechanisms in innate immune responses [29]. The activation of signal transduction pathways by Tlrs leads to important host defense functions, including inflammatory cytokines, chemokines, major histocompatibility complexes, and co-stimulatory molecules. Type I interferon plays a crucial role in antiviral infections [30]. Tlr signaling pathways were also analyzed. The expression levels of Ifna2, Ifna4, Ifna5, Tlr3, Ccl12, and Tlr9 in the Tlr signaling pathway were up-regulated in the rRC-HL-4d, CVS-24-4d, and GX074-7d groups. The up-regulated genes at 7 dpi were Tnf, Ikbke, Cxcl10, MyD88, and Ccl5 (Figure 6b). MyD88 potentially interacts with TRAF3 and Tlr9, with scores of 0.998 and 0.999, respectively (Appendix A).

We then performed Pearson correlation coefficient analyses on selected genes. Stat1, Sphk1, and viperin showed significant positive correlations (*p* < 0.05), as did MyD88, MCP-1 (Ccl2), Tnf-α, Il-1β, lgtp, Stat1, and Ifn-β. Viperin demonstrated higher expression levels in CVS-24-7d and GX074-7d. MCP-1 also showed a trend towards higher expression in rRC-HL-4d and CVS-24-4d (Appendix A).

## 4. Discussion

Rabies is one of the oldest zoonoses and has been described as one of the most terrifying diseases known to humans. Once rabies symptoms appear, the vast majority of infections are fatal. Every year, 70,000 people are killed by rabies worldwide [1,2]. However, little is known about how RABV eludes the immune response and kills hosts. Inflammation plays an important role in RABV, as demonstrated by the activation of microglia and astrocytes. In this study, we explored whether rabies is related to brain microglia or not. We conducted an mRNA-Seq transcriptomics analysis of microglia isolated by MACS from the brains of mice infected with RABV strains of varying virulence.

As shown in Figure 1, we used CD11b microglia microbeads to isolate and dissociate the single microglia from the brain (Figure 1). The purity of microglial cells in the positive fraction was 88.39%. The minimum survival rate of microglia was 81.91% at 4 dpi in mice infected with GX074, and the maximum survival rate was 96.7% in mice at 7 dpi infected with rRC-HL. Thus, this method is appropriate for studying microglia in brains infected with RABV.

In previous studies, Fu et al. found that attenuated RABV could activate type I interferon signaling pathways and inflammatory chemokines in the mouse brain and the mitochondrial antiviral signaling pathway, inducing cytokine expression in astrocytes. Conversely, a street RABV strain evaded the host innate immune system [31]. Immune responses and inflammation differ between attenuated and street rabies viruses. A recent study revealed that street RABV infection induced the upregulation of some chemokine expression levels and activated MAPK and NF-κB signaling pathways in dog, human, and mouse brain tissue [32]. We compared transcriptome profiles induced by different RABV strains: the attenuated rRC-HL, the standard challenge virulent strain CVS-24, and the street strain GX074. We found that many genes in the brain were upregulated as the disease worsened (Figure 2c). In addition, many signaling pathways were activated (Figure 5c,d). The NF-κB and MAPK signaling pathways were upregulated in DMEM vs. GX074 at 4 and 7 dpi in the microglia. Other RABV strains presented varying degrees of changes in the NF-κB and MAPK signaling pathways, implying the importance of microglia in regulating the NF-κB and MAPK signaling pathways. Moreover, the Tlr, Tnf, RIG-I, NOD, and Jak-Stat signaling pathways were highly activated in microglia by RABV at 4 and 7 dpi. These results showed that RABV can activate many pathways related to inflammation and immunity in the microglia, and microglia are involved in the regulation of inflammation and the innate immune system.

RABV affects innate immune signaling within infected cells. Lena Feige found that M and P proteins in RABV mediated the inhibition of innate immune gene expression [33]. However, RABV has developed mechanisms to suppress broader inflammatory response. RABV can naturally replicate in infected humans and animals, but there is a lack of significant infiltration by inflammatory immune cells. Moreover, street RABV evades the host immune response by restricting the expression of the G protein in virions [34,35], and the interaction of the P protein of RABV with STAT proteins plays a pivotal role in lethal rabies disease [36].

rRC-HL, GX074, and CVS-24 induced 3622 and 4590, 265 and 4901, and 4079 and 6337 DEGs at 4 and 7 dpi, respectively. In addition, the up-regulated DEG numbers compared to DMEM groups at 4 and 7 dpi in mice infected with rRC-HL, GX074, and CVS-24 were 1953 and 2557, 158 and 2667, and 2302 and 3091. The numbers of differentially expressed genes (DEGs) were up-regulated in comparison to the DMEM group at 4 and 7 dpi in mice infected with rRC-HL, GX074, and CVS-24, reaching 1669 and 2033, 102 and 2234, and 2302 and 1795, respectively (Appendix A). Based on the above data, although the street strain GX074 is the most virulent RABV strain and causes typical neurological symptoms leading to death, it activated the fewest DEGs at 4 dpi. However, DEGS in GX074-7d are higher than in rRC-HL-7d and lower than in CVS-24-7d. This phenomenon may be related to immune evasion, causing P, M, and G proteins in GX074 to induce fewer DEGs at 4 dpi. Later, GX074-7d produced more DEGs or activated immune pathways. Therefore, this phenomenon deserves further mechanistic studies.

The upregulation of inflammatory cytokines and chemokines in the attenuated high-egg-passage Flury (HEP-Flury) strain and virulent challenge virus standard-11 (CVS-11) strains infected mice may represent interactions among neurons, astrocytes, microglia, and even infiltrating T cells [37]. Our findings demonstrate that microglia are involved in the upregulation of inflammatory cytokines and chemokines. The DEGs in microglia identified by transcriptomics analyses in this study included the response to stress, activation of innate and adaptive immune pathways, up-regulation of inflammatory cytokines and chemokines, and changes in metabolites, indicating that microglia play a major role in immune and inflammatory metabolism.

Most neurons in the brain and spinal cord of patients infected with the street rabies viruses are intact and have few abnormalities. Conversely, fixed rabies viruses (CVS11) can cause widespread damage to neurons in the brain and spinal cord, apoptosis, and inflammation. This difference is probably related to the mechanism of neuronal dysfunction, the mode of viral spread in the brain, and the nature of the stimulus for inflammatory infiltration [38,39,40]. Inflammatory response (pathway) was strong in CVS-24 at 4 dpi, however, GX074 was low at 4 dpi (Figure 5c). It is the different strains that lead to the results.

The activation of PRRs by pathogenic or endogenous motifs initiates innate immunity. Innate immune cells express high levels of PRRs and Tlrs [32]. Tlrs activate signaling pathways dependent on the adaptors MyD88 or TRIF to trigger innate immune responses and then induce the release of proinflammatory cytokines, type I interferon, chemokines, and antimicrobial proteins [30]. Tnf-α is released by microglial cells after activation of Tlr3 and Tlr4, demonstrating that Tlrs influence Tnf-α [41]. In this study, cytokines such as Tnf, Cxcl10, MyD88, and Ccl5 were increased in infected groups, suggesting that RABV activated Tlr and Tnf signaling pathways to induce a greater expression of proinflammatory cytokines, type I interferon, and chemokines to resist RABV.

## 5. Conclusions

We successfully dissociated single microglia from the brain by MACS. RABV caused significant changes in the mRNA expression profiles in microglia from infected mouse brains. These transcriptomic data explain the molecular mechanisms of microglia infected with RABV through the upregulation of inflammatory cytokines, chemokines, and metabolites to resist RABV and regulate the Tlr, Tnf, RIG-I, NOD, NF-κB, MAPK, and Jak-STAT signaling pathways. These results provide important information for studying RABV pathogenesis and immune mechanisms.

## Data Availability

All data generated or analyzed during this study are available in this article and the supplementary information files. The transcriptomic raw data presented in the study are deposited in Figshare: https://doi.org/10.6084/m9.figshare.22586284 (accessed on 12 March 2023).

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
