# Peer review of "Transcriptomic Analysis of mRNA Expression Profiles in the Microglia of Mouse Brains Infected with Rabies Viruses of Varying Virulence"

_viruses, 2023, doi:10.3390/v15061223_

Round 1

Reviewer 1 Report

This manuscript explored the mRNA expression profiles in the microglia of mouse brains intracerebrally infected with RABV. This study will helpful for revealing the pathogenic mechanism in RABV infected microglial.

1.    Please consider moving Table 1 of results to the Materials and Methods section as it would be more appropriate there instead of in the results section.

2.    There are many mistakes in the article, such as the format is wrong. the singular and plural numbers of nouns are inconsistent. such as line 41- and line 30-. Commas are double in line 90. There is no comma in line 88. It is necessary to correct these mistakes in the paper.

3.    Although I have tried to incorporate/highlight corrections of the language, sentences and spellings i.e., it is hard to understand at some places. I would suggest authors to revise the text with some professional help.

4.    Line 129- and line 130- should be moved to line 132-.

5.    DMEM-4d vs. rRC-HL-4d, DMEM-4d vs GX074-4d and DMEM vs. CVS-24 should be displayed in Venn diagram of Fig.3 except for rRC-HL-4d vs CVS-24-4d and rRC-HL-4d vs GX074-4d.

6.    Please discuss up and down-regulated DEGs separately.

7.    Authors should add the detail method of clinical score of infected mice.

Minor editing of English language required

Author Response

Dear Dr. Luo,

Thank you again for your manuscript submission: Manuscript ID: viruses-2371275

Title: Transcriptomic analysis of mRNA expression profiles in the microglia
of mouse brains infected with rabies viruses of varying virulence

Dear Dr.  Eric O. Freed, Editor-in-Chief, Viruses

We are grateful to you and your reviewers so much for your useful and valuable advices and impetus on our manuscript. According to your suggestions, we described the necessary points one by one to respond to your concerns as indicated in this response letter.

This manuscript is edited by LetPub (ACCDON Ltd Co., USA) and a biological scientist, a speaker of native English.

We now re-submitted our revised manuscript “Transcriptomic analysis of mRNA expression profiles in the microglia of mouse brains infected with rabies viruses of varying virulence” (Manuscript ID: viruses- 2371275) to Viruses.

We are hopeful that the revision is satisfactory to you as well as the reviewers and be accepted. We are looking forward to hearing from you soon.

We deeply appreciate you and your reviewers again for your constructive and insightful suggestions to our manuscript.

Sincerely yours

Ting Rong Luo

Phone: +086-771-3232248;

Email:  tingrongluo@gxu.edu.cn

Response letter

Reviewer #1:

General comments:

This manuscript explored the mRNA expression profiles in the microglia of mouse brains intracerebrally infected with RABV. This study will be helpful in revealing the pathogenic mechanisms of RABV infected microglial cells.

Specific comments:

Point 1: Please consider moving Table 1 of results to the Materials and Methods section as it would be more appropriate there instead of in the results section.

Response 1: We agree with reviewer’s suggestion. We have moved the Table 1 from the Results section to the Materials and Methods section in Lines 185.

The original paragraph is: 

RT-qPCR primers were designed using sequences obtained from NCBI databases (https://www.ncbi.nlm.nih.gov/). The primers were synthesized and purified by BGI

The changed paragraph is:

The twelve primer pairs (Table 1) were designed and obtained from NCBI databases (https://www.ncbi.nlm.nih.gov/). The primers were synthesized and purified by BGI.

Point 2: There are many mistakes in the article, such as the format is wrong. the singular and plural numbers of nouns are inconsistent. such as line 41- and line 30-. Commas are double in line 90. There is no comma in line 88. It is necessary to correct these mistakes in the paper.

Response 2: We agree with reviewer’s suggestion. We have corrected the singular and plural numbers of nouns in line 40.  We removed a comma from line 90 and added a comma in line 88. And we have corrected some mistakes in the other places of the article. Detailed information is in the paper.

The original paragraph is: 

line 38: The DEGs were evaluated using GO, KEGG and PPI network analyses.

The changed paragraph is:

line 38: The DEGs were evaluated using GO, KEGG and PPI network analysis.

The original paragraph is: 

Line 98-99: and that of CVS-24 was 10−5.843/ml according to Reed-Muench[19]..

The changed paragraph is:

line 98-99: and that of CVS-24 was 10−5.843/ml according to Reed-Muench[19].

The original paragraph is: 

Line 97: The titers of rRC-HL were determined as 3.967 × 107 focus forming unit per milliliter (FFU/mL) in N2a cells by IF The 50% lethal dose (LD50) of GX074

The changed paragraph is:

Line 97: The titers of rRC-HL were determined as 3.967 × 107 focus forming unit per milliliter (FFU/mL) in N2a cells by IF. The 50% lethal dose (LD50) of GX074

Point 3: Although I have tried to incorporate/highlight corrections of the language, sentences and spellings i.e., it is hard to understand at some places. I would suggest authors to revise the text with some professional help.

Response3: Thank you for reviewer’s suggestion. The manuscript was revised by professional help. In fact, this manuscript is edited by LetPub (ACCDON Ltd Co., USA) a biological scientist and a native English speaker. We have corrected spellings and sentences errors in the article. Detailed information are in the paper. Please check it.

Point 4: Line 129- and line 130- should be moved to line 132-.

Response 4: We agree with reviewer’s suggestion. The three references are very valuable and we have cited them in Lines 67-73 as references [15-17].

The original paragraph is:

Line 145: separator to a suitable tube and flushed with PBS to collect the purified microglia population. Microglia purity was assessed by IF and flow cytometry.

The changed paragraph is:

Line 147: Microglia purity was assessed by IF and flow cytometry. Microglial cells were fluorescently stained with CD11b- PE (BD, USA, NJ,553311) and CD45-FITC (BD, USA, NJ,553080). Cell debris and dead cells were excluded from the analysis.

Point 5: Line78: DMEM-4d vs. rRC-HL-4d, DMEM-4d vs GX074-4d and DMEM vs. CVS-24 should be displayed in Venn diagram of Fig.3 except for rRC-HL-4d vs CVS-24-4d and rRC-HL-4d vs GX074-4d.

Response 5: Thank you for reviewer’s suggestion. DMEM-4d vs rRC-HL-4d, DMEM-4d vs GX074-4d and DMEM vs CVS-24 are based on DMEM-4d to analyze, but rRC-HL-4d vs CVS-24-4d and rRC-HL-4d vs GX074-4d are rRC-HL-4d. They are analyzed in two aspects.  

Point 6: Please discuss up and down-regulated DEGs separately.

Response 6: We agree with reviewer’s suggestion. We have discussed up and down-regulated DEGs separately by a table in supplementary materials. Please check it in line 493-497.

Line 493-497: In addition, the up-regulated DEG numbers compared to DMEM groups at 4 and 7 dpi in mice infected with rRC-HL, GX074, and CVS-24 were 1953 and 2557, 158 and 2667, and 2302 and 3091.the numbers of differentially expressed genes (DEGs) were up-regulated in comparison to the DMEM group at 4 and 7 dpi in mice infected with rRC-HL, GX074, and CVS-24, reaching 1669 and 2033, 102 and 2234, and 2302 and 1795, respectively (Supplementary Table 2

Point 7: Authors should add the detail method of clinical score of infected mice.

Response 7: We agree with reviewer’s suggestion. We have added the detailed method to assess clinical score of infected mice. Please see the revised paragraph in Lines 103-104, Lines 501-502 and Lines 507-509.

The paragraph is:

Line 117-118: Disease progression was evaluated by scoring clinical signs and body weight change as previously described[20].

[20] Chopy D, Pothlichet J, Lafage M, Mégret F, Fiette L, Si-Tahar M, et al. Ambivalent role of the innate immune response in rabies virus pathogenesis. J Virol. 2011; 85: 6657–68.

Line 262-263: The clinical scores were assessed daily for 7 days. Body weight changes and clinical scores were evaluated by two-way ANOVA test, *p<0.05, **p<0.01.

Line 270-273: Mouse clinical score: 0, normal; 1, disorderly hair and slackness; 2, depressed spirit, loss of appetite, hunched back, and sensitivity to stimulation; 3, tremors and paralysis of the fore or hind limbs; 4, hypothermia, abundant white discharge from the eyes, and respiratory failure, and 5, death.

Reviewer 2 Report

Review of Liu et al Transcriptomic analysis of mRNA expression profiles in the microlia of mouse brains infected with rabies viruses of varying virulence

Line 117. How were they challenged?  IM? The initial inoculation was IC but did not see more information about the challenge inoculation.  The criteria for euthanasia is incredibly vague.  What were the neurological signs that required euthanasia?  Was wt loss included in the euthanasia criteria? How were mice euthanized?

Line 125.  Several times “varying virulence” is stated.  Please include a table of the strains used and virulence.  rRC-HL is a vaccine strain but what is it’s origin?  Dog?

Were parts of the microglia from each part of the brain isolated individually? Or were the microglia from all areas combined for staining?

Line 225-226. Please elaborate on your viruses.  Where any passaged in cell culture multiple times?  Why inoculate IC vs IM or SC, which would be a more traditional route of human infection?

Line 239.  What was your % weight loss that required euthanasia per your IACUC?  Please include numbers and not %. 

Line 245 Why weren’t mice that “showed typical neurological symptoms at 5dpi”  euthanized as required per your methods?

Line 250.  What does “A slight depressed spirit” mean?

Figure 1.  Is there a way to improve the micrographs eg a counter stain or some way to decrease what appears as oversaturation?  Or maybe it’s just my printer…….

Line 263 and 264.  The scoring should have been included in methods.   Were animals that scored a 3 euthanized per the neurological signs euthanasia requirement?

Line 272.  Can you estimate the number of cells that may have been lost just through basic dissection and processing?

Line 288. Is “to” missing between “conducted” and “further”

Figure 2C.  Why wasn’t the DMEM data included in this figure?

Line 378 Why were the mice allowed to die vs euthanizing when neurological signs were present?

Figure 5.  What does it mean if the space is blank?  The comparison wasn’t done or no difference?  Eg Fig 5c DMEM 4d vs GX074 there are several missing dots.

Discussion

There are several sentences that are not cited, esp in the first two paragraphs

It appears that the DMEM control and the dog variant are less variable, likely since it is a wild type virus and wants to downregulate inflammation etc.. similar to some bat variants

 Please include discussion about CVS being a murine cell culture adapted variant which would help explain its grown in mice vs a street variant that had not been grown in murine cell culture. 

Are you surprised by the DMEM inflammatory response wasn’t greater since it was directly inoculated into the brains of animals?

There have been several papers about the immune response in the brains of animals, include many mouse studies.  How is yours unique, beyond the ability to dissociate individual microglia?

English quality acceptable, minor check required

Author Response

Dear Dr. Luo,

Thank you again for your manuscript submission: Manuscript ID: viruses-2371275

Title: Transcriptomic analysis of mRNA expression profiles in the microglia
of mouse brains infected with rabies viruses of varying virulence

Dear Dr.  Eric O. Freed, Editor-in-Chief, Viruses

We are grateful to you and your reviewers so much for your useful and valuable advices and impetus on our manuscript. According to your suggestions, we described the necessary points one by one to respond to your concerns as indicated in this response letter.

This manuscript is edited by LetPub (ACCDON Ltd Co., USA) and a biological scientist, a speaker of native English.

We now re-submitted our revised manuscript “Transcriptomic analysis of mRNA expression profiles in the microglia of mouse brains infected with rabies viruses of varying virulence” (Manuscript ID: viruses- 2371275) to Viruses.

We are hopeful that the revision is satisfactory to you as well as the reviewers and be accepted. We are looking forward to hearing from you soon.

We deeply appreciate you and your reviewers again for your constructive and insightful suggestions to our manuscript.

Sincerely yours

Ting Rong Luo

Phone: +086-771-3232248;

Email:  tingrongluo@gxu.edu.cn

Response letter

Reviewer #2:

Major comments: Review of Liu et al Transcriptomic analysis of mRNA expression profiles in the microglia of mouse brains infected with rabies viruses of varying virulence

Point 1: Line 117. How were they challenged?  IM? The initial inoculation was IC but did not see more information about the challenge inoculation.  The criteria for euthanasia is incredibly vague.  What were the neurological signs that required euthanasia?  Was wt loss included in the euthanasia criteria? How were mice euthanized?

Response 1:  Thank you for reviewer’s suggestion. Firstly, Mice were challenged by IC. The detailed results are in line 112-115. For the virus-infected group, mice were inoculated intracerebrally with 1000 ffu of rRC-HL in 30 µl DMEM. The mice were challenged with CVS-24 and GX074 at 100 LD50 in 30 µl DMEM, and the control group was inoculated with 30 µl DMEM. Secondly, we collected mouse brain at 4 dpi and 7 dpi. Mice in the infected groups start to show clinical symptoms at 4 dpi, the neurological signs (tremors and paralysis of the fore or hind limbs, hypothermia, abundant white discharge from the eyes) are most obvious at 7 dpi. Mice infected with CVS-24 showed body weight loss (3 dpi), ruffling of hair, loss of coordination (4-5 dpi), tremors, paralysis of the fore or hind limbs and hind limb palsy (6-7 dpi). The mice infected with GX074 tended to show weight loss, mania, convulsions, trembling, shaking, and even paralysis. The mice infected with rRC-HL showed a slight weight loss (4 dpi) and depression in spirit (5-6 dpi), and are recovered at 7 dpi. So we changed the sentence to a clear expression.

The original paragraph is: Line 119-120: The mice with typical neurological symptoms of rabies were euthanized.

The changed paragraph is: Line119-120: The mice were then euthanized at 4 dpi and 7 dpi using halothane in a closed container.

Point 2:   Line 125.  Several times “varying virulence” is stated.  Please include a table of the strains used and virulence.  rRC-HL is a vaccine strain but what is it’s origin?  Dog?

Were parts of the microglia from each part of the brain isolated individually? Or were the microglia from all areas combined for staining?

Response 2: Thank you for reviewer’s suggestion. The titers in strains have been described in line 87-89 of 2.2 Virus titration

The changed paragraph is: The titers of rRC-HL were determined as 3.967 × 107 focus forming unit per milliliter (ffu/ml) in N2a cells by IFA. The 50% lethal dose (LD50) of GX074 was calculated as 10−4.898/ml  to adult mice[18] and that of CVS-24 was 10−5.843/ml according to Reed-Muench[19].

Whether we make a table of virus titers?

“rRC-HL is a vaccine strain but what is it’s origin?  Dog?”

Response: The RC-HL strain was established from the Nishigahara strain which had been maintained by rabbit brain passages, after 294 passages in chicken embryos, 8 passage in chicken embryo fibroblasts (CEF), 5 passages in Vero cells and 23 passages in hamster lung (HmLu) cells [1].

[1] Ito N, Kakemizu M, Ito KA, Yamamoto A, Yoshida Y, Sugiyama M, et al. A comparison of complete genome sequences of the attenuated RC-HL strain of rabies virus used for production of animal vaccine in Japan, and the parental Nishigahara strain. Microbiol Immunol. 2001; 45:51–8

“Were parts of the microglia from each part of the brain isolated individually? Or were the microglia from all areas combined for staining?”

Response: The sorted microglia are isolated from the whole brain of mice individually.

“Or were the microglia from all areas combined for staining?”

Response: Yes.

Point 3:  Line 225-226. Please elaborate on your viruses.  Where any passaged in cell culture multiple times?  Why inoculate IC vs IM or SC, which would be a more traditional route of human infection?

Response3: We agree with reviewer’s suggestion. Virus passages have been introduced in line 96-99. To IC, IM and SC, IC and IM would be a common infection route, but IM would be a more traditional route of human infection.

The changed paragraph is : Line 96-99:The titers of rRC-HL in 6 passage were determined as 3.967 × 107 focus forming unit per milliliter (ffu/ml) in N2a cells by IFA. The 50% lethal dose (LD50) of GX074 in 6 passage was calculated as 10−4.898/ml in adult mice[18] and that of CVS-24 in 8 passage was 10−5.843/ml according to Reed-Muench[19].

Point 4: Line 239.  What was your % weight loss that required euthanasia per your IACUC?  Please include numbers and not %. 

Response 4: Thank you for reviewer’s suggestion. We collected mouse brain at 4 dpi and 7 dpi. Mice in infected groups start to show symptom at 4 dpi, the neurological signs (tremors and paralysis of the fore or hind limbs, hypothermia, abundant white discharge from the eyes) are most obvious at 7 dpi, which is our purpose to study. Most importantly, our animal experiments were conducted according to the guidelines of the National Institute of Health Guide for the Care and Use of Laboratory Animals, and the experimental protocols were approved by the Ethical Committee of Guangxi University (GXU2019-021), Guangxi, China. All virus experiments were performed in microbiological safety cabinets in a Biosafety Level 3 laboratory. All husbandry procedures were conducted in accordance with the Animal Welfare Act and the Guide for the Care and Use of Laboratory Animals.

Point 5: Line 245 Why weren’t mice that “showed typical neurological symptoms at 5dpi” euthanized as required per your methods?

Response 5: Thank you for reviewer’s suggestion. 4-weeks- old Kunming mice, its body weight about 18-20 grams were used in our experiments. We collected mouse brain at 4 dpi and 7 dpi, respectively. Mice in the infected groups (CVS-24 and GX074) start to show symptoms at 4 dpi, the neurological signs (tremors and paralysis of the fore or hind limbs, hypothermia, abundant white discharge from the eyes) appear at 7 dpi. So we collect mouse brains at 4 dpi to euthanize.

Point 6: Line 250.  What does “A slight depressed spirit” mean?

Response 6: The sentence means that the mice injected with rRC-HL showed a slight depression in spirit.

The original paragraph is:

Line 241: showing disorderly hair, slackness, and a slight depression in spirit (Fig.1b).

The changed paragraph is:

Line 241: showing disordered hair, slackness, and a slight depression.

Point 7: Figure 1.  Is there a way to improve the micrographs eg a counter stain or some way to decrease what appears as oversaturation?  Or maybe it’s just my printer…….

Response 7: Thank you for your suggestion. We are so sorry that the picture is vague. We have adjusted the contrast as much as possible to make the picture clear. Please check it.

Point 8:  Line 263 and 264.  The scoring should have been included in methods.   Were animals that scored a 3 euthanized per the neurological sign euthanasia requirement?

Response 8: Thank you for reviewer’s suggestion. We have written scoring in Line 108 of “2.3 Mouse infection test”.

The original paragraph is: 

Disease progression was evaluated by scoring clinical signs and body weight change as previously described[20].

[20] Chopy D, Pothlichet J, Lafage M, Mégret F, Fiette L, Si-Tahar M, et al. Ambivalent role of the innate immune response in rabies virus pathogenesis. J Virol. 2011; 85: 6657–68.

“Were animals that scored a 3 euthanized per the neurological signs euthanasia requirement?”

Response: Thank you for reviewer’s suggestion. Animals were not scored a 3 euthanized per the neurological sign euthanasia requirement. we collected mouse brain and euthanized at 4 dpi and 7 dpi. The details are in line 224-232.

Line 224-232: The rabies symptoms of CVS-24-infected mice appeared at 4 dpi, and typical neurological symptoms were observed at 6 and 7 dpi; the highest clinical score was four, presenting hypothermia, abundant white discharge from the eyes, and respiratory failure. The mice infected with GX074 showed typical neurological symptoms at 5 dpi, which became severe at 7 dpi, presenting as a clinical score ranging from three to four at 6 and 7 dpi, tremors, and paralysis of the fore or hind limbs. In contrast, the mice infected with rRC-HL had mild symptoms, occurring from 5 to 7 dpi with a score of one to two in the broken-line graph. So mice are euthanized according days and symptom.

Point 9: Line 272.  Can you estimate the number of cells that may have been lost just through basic dissection and processing?

Response 9:  We are so sorry that we cannot estimate the number of cells that may have been lost just through basic dissection and processing, we estimated the total number of live microglia (3.345x105-3.45x106 cells, data not shown.) from a brain of mouse by cell counting, which is consistent to this paper.

 Hui Z, Rei S, Tomiuk S, et al. Efficient isolation of viable primary neural cells from adult murine brain tissue based on a novel automated tissue dissociation protocol. 2016.

Point 10:  Line 288. Is “to” missing between “conducted” and “further”

Response 10: Thank you for your suggestion.  It is careless to miss “to” between “conducted” and “further”. We have added it in line 278-279. Please check it.

The original paragraph is:

The paragraph changed to: mRNA-seq was conducted further understand the effects of RABV infection on the microglia transcriptome.

The changed paragraph is:

The paragraph changed to: mRNA-seq was conducted to further understand the effects of RABV infection on the microglia transcriptome.

Point 11:   Figure 2C.  Why wasn’t the DMEM data included in this figure?

Response 11: It is kind of reviewer and thank you for your suggestion. All the groups are compared with DMEM group, so DMEM data was not included in this figure 2c.

Point 12: Line 378 Why were the mice allowed to die vs euthanizing when neurological signs were present?

Response 12: Thank you for your kind reminding. we collected mouse brain and euthanized at 4 dpi and 7 dpi. Most of Mice in infected groups (CVS-24/GX074) start to show symptoms at 4 dpi, the neurological signs (tremors and paralysis of the fore or hind limbs, hypothermia, abundant white discharge from the eyes) are most obvious at 7 dpi. So when neurological signs were present at 4 dpi, mice were euthanized.

Point 13: Figure 5.  What does it mean if the space is blank?  The comparison wasn’t done or no difference?  Eg Fig 5c DMEM 4d vs GX074 there are several missing dots.

Response 13: Thank you for reviewer’s question and suggestion. “The Blank space” means there are no enriched pathways in comparison group. 

Point 14: Discussion

There are several sentences that are not cited, esp in the first two paragraphs

It appears that the DMEM control and the dog variant are less variable, likely since it is a wild type virus and wants to downregulate inflammation etc., similar to some bat variants

 Please include discussion about CVS being a murine cell culture adapted variant which would help explain its grown in mice vs a street variant that had not been grown in murine cell culture. 

Are you surprised by the DMEM inflammatory response wasn’t greater since it was directly inoculated into the brains of animals?

There have been several papers about the immune response in the brains of animals, include many mouse studies.  How is yours unique, beyond the ability to dissociate individual microglia?

“There are several sentences that are not cited, esp in the first two paragraphs”

Response 14: Thank you for reviewer’s question and suggestion, and we agreed with reviewer. We have added citation in line 452-454.

The original paragraph is:

Line 452-454: Rabies is one of the oldest zoonoses and has been described as one of the most terrifying diseases known to humans. Once rabies symptoms appear, the vast majority of infections are fatal. Every year, 70,000 people are killed by rabies worldwide.

The changed paragraph is:

Line 452-454: Rabies is one of the oldest zoonoses and has been described as one of the most terrifying diseases known to humans. Once rabies symptoms appear, the vast majority of infections are fatal. Every year, 70,000 people are killed by rabies worldwide [1-2].

[1] Fooks AR, Cliquet F, Finke S, Freuling C, Hemachudha T, Mani RS, et al. Rabies. Nat Rev Dis Primer. 2017;3:17091.

[2] Fooks AR, Banyard AC, Horton DL, Johnson N, McElhinney LM, Jackson AC. Current status of rabies and prospects for elimination. The Lancet. 2014;384:1389–99.

“It appears that the DMEM control and the dog variant are less variable, likely since it is a wild type virus and wants to downregulate inflammation etc.. similar to some bat variants”

Response: Thank you for reviewer’s question. DMEM control is negative control and stable. DMEM is injected intracerebrally into the brain, maybe causing mild temporary inflammation. However, dog variant would cause inflammation during infection process, but inflammation is downregulated, which is similar to bat variants.

“Please include discussion about CVS being a murine cell culture adapted variant which would help explain its grown in mice vs a street variant that had not been grown in murine cell culture.”

Response: Thank you for reviewer’s question and we are agreed with reviewer. And we have discussed CVS in discussion. The details are in line 515-521.

Line 515-521: Most neurons in the brain and spinal cord of patients infected with street viruses are intact and have few abnormalities. On the other hand, fixed viruses (CVS11) is the opposite, which can cause widespread damage to neurons in the brain and spinal cord, apoptosis and inflammation. This difference is probably related to the mechanism of neuronal dysfunction, the mode of viral spread in the brain and the nature of the stimulus for inflammatory infiltration[38–40]. Inflammatory response(pathway) was strong in CVS-24 at 4 dpi, however, GX074 was few at 4 dpi (Fig.5c). It is the different strains that lead to the results.

[38]         Kojima, D.; Park, C.-H.; Satoh, Y.; Inoue, S.; Noguchi, A.; Oyamad, T. Pathology of the Spinal Cord of C57BL/6J Mice Infected with Rabies Virus (CVS-11 Strain). J. Vet. Med. Sci. 2009, 71, 319–324, doi:10.1292/jvms.71.319.

[39]  Kojima, D.; Park, C.-H.; Tsujikawa, S.; Kohara, K.; Hatai, H.; Oyamad, T.; Noguchi, A.; Inoue, S. Lesions of the Central Nervous System Induced by Intracerebral Inoculation of BALB/c Mice with Rabies Virus (CVS-11). J. Vet. Med. Sci. 2010, 72, 1011–1016, doi:10.1292/jvms.09-0550.

[40]  Miyamoto, K.; Matsumoto, S. COMPARATIVE STUDIES BETWEEN PATHOGENESIS OF STREET AND FIXED RABIES INFECTION. J. Exp. Med. 1967, 125, 447–456.

“Are you surprised by the DMEM inflammatory response wasn’t greater since it was directly inoculated into the brains of animals?

Response: No. Inflammatory response wasn’t greater in DMEM group, showing that our control group is correct. DMEM is composed of amino acids and glucose. DMEM is injected intracerebrally into the brain, causing mild temporary inflammation. However, RABV infection could cause inflammation at 4 dpi and 7 dpi. Our results showed that infected groups (rRC-HL, GX074   and CVS-24) have obvious inflammatory response.

“There have been several papers about the immune response in the brains of animals, include many mouse studies.  How is yours unique, beyond the ability to dissociate individual microglia?”

Response: Thank you for reviewer’s question. There are so many transcriptomic analyses in brains, BV2, also other nerve cells [8]. As we can see in line 64-68: Kim et al. reported early transcriptional changes in vitro in the BV2 microglial cell line and neurons infected with RABV [11]. We successfully dissociated single microglia from the brain by MACS in adult mouse and conducted transcriptomic analysis. Moreover, our results are more comprehensive. First, we collect samples at 4 dpi and 7 dpi. second, we compare DMEM, rRC-HL and GX074, showing that our results are unique.

[8] Sui B, Chen D, Liu W, Tian B, Lv L, Pei J, et al. Comparison of lncRNA and mRNA expression in mouse brains infected by a wild-type and a lab-attenuated Rabies lyssavirus. J Gen Virol. 2021;102

[11] Kim S, Larrous F, Varet H, Legendre R, Feige L, Dumas G, et al. Early transcriptional changes in rabies virus-infected neurons and their impact on neuronal functions. Front Microbiol. 2021;12:730892

Reviewer 3 Report

Microglia, as immune effector cells resident in the central nervous system, is key cells for the maintenance of the entire brain. They constantly remove plaques, damaged, unnecessary neurons, synapses, and infectious factors in the central nervous system. The study of microglia after RABV infection has a good guiding significance for the treatment of rabies. In this experiment, transcriptome sequencing was performed to detect the changes of transcription levels of microglia-related factors in mouse brain tissue after RABV infection, and the differences in inflammatory factors and immune-related factors in mouse microglia after infection with strong and weak strains were discussed. The research is interesting. In the preface, the relevant research background is supplemented. The discussion part is too simple. The authors should combine the presented data and other researches to explain the changes of related pathways in microglia caused by viruses, and put forward constructive research opinions. 

Author Response

Dear Dr. Luo,

Thank you again for your manuscript submission: Manuscript ID: viruses-2371275

Title: Transcriptomic analysis of mRNA expression profiles in the microglia
of mouse brains infected with rabies viruses of varying virulence

Dear Dr.  Eric O. Freed, Editor-in-Chief, Viruses

We are grateful to you and your reviewers so much for your useful and valuable advices and impetus on our manuscript. According to your suggestions, we described the necessary points one by one to respond to your concerns as indicated in this response letter.

This manuscript is edited by LetPub (ACCDON Ltd Co., USA) and a biological scientist, a speaker of native English.

We now re-submitted our revised manuscript “Transcriptomic analysis of mRNA expression profiles in the microglia of mouse brains infected with rabies viruses of varying virulence” (Manuscript ID: viruses- 2371275) to Viruses.

We are hopeful that the revision is satisfactory to you as well as the reviewers and be accepted. We are looking forward to hearing from you soon.

We deeply appreciate you and your reviewers again for your constructive and insightful suggestions to our manuscript.

Sincerely yours

Ting Rong Luo

Phone: +086-771-3232248;

Email:  tingrongluo@gxu.edu.cn

Response letter

Reviewer #3:

comments:

Comments and Suggestions for Authors

Microglia, as immune effector cells resident in the central nervous system, is key cells for the maintenance of the entire brain. They constantly remove plaques, damaged, unnecessary neurons, synapses, and infectious factors in the central nervous system. The study of microglia after RABV infection has a good guiding significance for the treatment of rabies. In this experiment, transcriptome sequencing was performed to detect the changes of transcription levels of microglia-related factors in mouse brain tissue after RABV infection, and the differences in inflammatory factors and immune-related factors in mouse microglia after infection with strong and weak strains were discussed. The research is interesting. In the preface, the relevant research background is supplemented. The discussion part is too simple. The authors should combine the presented data and other researches to explain the changes of related pathways in microglia caused by viruses, and put forward constructive research opinions. 

Response: Thank you for reviewer’s question and we are agreed with reviewer.

Line 466-484: In the previous studies, Fu et al. found that attenuated RABV could activate type I interferon signaling pathways and inflammatory chemokines in the mouse brain and the mitochondrial antiviral signaling pathway, inducing cytokine expression in astrocytes. Conversely, a street RABV strain evaded the host innate immune system [27, 28]. Immune responses and inflammation differ between attenuated and street rabies viruses. A recent study revealed that street RABV infection induced the upregulation of some chemokine expression levels and activated MAPK and NF‑κB signaling pathways in dog, human, and mouse brain tissues[29]. We compared transcriptome profiles induced by different RABV strains; the attenuated rRC-HL, the standard challenge virulent strain CVS-24, and the street strain GX074. We found that many genes in the brain were upregulated as the disease worsened (Fig. 2c). Additionally, some signaling pathways were activated (Fig. 5c, 5d). The NF‑κB and MAPK signaling pathways were upregulated in DMEM vs GX074 at 4 and 7 dpi in the microglia. Other RABV strains presented varying degree of changes in the NF‑κB and MAPK signaling pathways, implying the importance of microglia in regulating NF‑κB and MAPK signaling pathways. Moreover, the Tlr, Tnf, RIG−I, NOD, and Jak−Stat signaling pathways were highly activated in microglia by RABV at 4 and 7 dpi.

These results showed that RABV can activate many pathways related to inflammation in the microglia, and microglia are involved in the regulation of inflammation and innate immune.

[27]  Martin, E.; El-Behi, M.; Fontaine, B.; Delarasse, C. Analysis of Microglia and Monocyte-Derived Macrophages from the Central Nervous System by Flow Cytometry. J Vis Exp 2017, 55781, doi:10.3791/55781.

[28]  Erny, D.; Dokalis, N.; Mezö, C.; Mossad, O.; Blank, T.; Prinz, M. Flow-Cytometry-Based Protocol to Analyze Respiratory Chain Function in Mouse Microglia. STAR Protoc 2022, 3, 101186, doi:10.1016/j.xpro.2022.101186.

[29]  Eyo, U.B.; Mo, M.; Yi, M.-H.; Murugan, M.; Liu, J.; Yarlagadda, R.; Margolis, D.J.; Xu, P.; Wu, L.-J. P2Y12R-Dependent Translocation Mechanisms Gate the Changing Microglial Landscape. Cell Reports 2018, 23, 959–966, doi:10/gddfmn.

Round 2

Reviewer 2 Report

Significant improvements have been made to this manuscript